# Prevalence of Periodontal Disease among Obese Young Adult Population in Saudi Arabia—A Cross-Sectional Study

**DOI:** 10.3390/medicina56040197

**Published:** 2020-04-24

**Authors:** Julie Toby Thomas, Toby Thomas, Masood Ahmed, Karthiga Kannan S, Zoha Abdullah, Sara Ayid Alghamdi, Betsy Joseph

**Affiliations:** 1Department of Preventive Dental Sciences, College of Dentistry, Majmaah University, Al Zulfi 11952, Saudi Arabia; 2Department of Restorative Dental Sciences, College of Dentistry, Majmaah University, Al Zulfi 11952, Saudi Arabia; 3Department of Maxillofacial Dental Sciences, College of Dentistry, Majmaah University, Al Zulfi 11952, Saudi Arabia; 4Department of Public Health Dentistry, Asan Memorial Dental College, Chengalpettu 6000094, India; 5Department of Preventive Dental Science, College of Dentistry, Majmaah University, Al Zulfi 11952, Saudi Arabia; 6Department of Periodontics and Community Dental Sciences, College of Dentistry, King Khalid University, Abha 62521, Saudi Arabia

**Keywords:** prevalence, periodontal disease, obese, CPI, Saudi Arabia

## Abstract

*Background and objectives*: We aimed to assess the prevalence of periodontal disease among obese young adults in Saudi Arabia and to analyze the association between different body mass indexes and the severity of periodontal disease. *Materials and methods*: This descriptive cross-sectional study consisted of 307 obese patients aged 18–39 years, with body mass index (BMI) ≥30. Demographic variables for periodontal disease, anthropometric parameters such as BMI along with clinical parameters such as oral hygiene index-simplified, community periodontal index (CPI) score and loss of attachment (LOA), were assessed. Multivariate binary logistic regression analysis was used to identify the predictors for chronic periodontitis in obese young adults between 18–40 years of age. *Results*: The majority of the participants (71.3%) had periodontal disease. Obese and extremely obese patients together showed a statistically significant difference in the age group of 21-30 years in terms of CPI score for inflammation (*p* < 0.05) and LOA (*p* < 0.001). Logistic regression analysis showed age (OR: 3.180; 95%CL: 1.337–7.561; *p* <.001), occasional dental visit (OR: 5.965; 95%CL: 3.130–11.368; *p* < 0.001), smoking >10 cigarettes (OR: 11.868; 95%CL: 3.588–39.254; *p* < 0.001) and poor oral hygiene status (OR: 17.250; 95%CL: 6.958–42.764; *p* < 0.001) were associated with a significantly higher risk of having periodontal disease. *Conclusions*: This study showed a high prevalence of periodontal disease in obese patients among the Saudi Arabian population.

## 1. Introduction

Obesity is now becoming an alarming global concern in developing countries, especially in Saudi Arabia. According to World Health Organization (WHO), obesity is defined an excess amount of body fat in proportion to body mass, resulting impairment of a patient’s overall health with BMI ≥30.0 kg/sq.m [1]. Prevalence of obesity among the younger adult population has been increasing in the modern era as a result of genetic susceptibility, sedentary lifestyles, poor dietary patterns, further reflecting the potential impact on morbidity and mortality of human health. Evidence-based studies have highlighted the harmful consequences of obesity amongst adolescents and young adults with a mean age of 22.4 years, out of which some of the most important concerns include systemic diseases like cardiovascular, type 2 diabetes mellitus, atherosclerosis, cancer, and osteoarthritis [2,3]. Certain studies have reported the possible role of bioactive molecules like adipocytokines released by adipocytes to activate monocytes, to trigger the production of inflammatory cytokines, which has a significant role in initiating periodontal disease. It is reported that an increase in tumor necrosis factor-α (TNF-α) levels may suppress the insulin-induced tyrosine phosphorylation of insulin receptor substrate-1, thereby blocking the translocation of glucose-transporting proteins, impairing insulin action [4].

Periodontal disease is an infectious and inflammatory disorder of tooth-supporting structures resulting from the interaction between pathogenic bacteria and the host immune response. A study done on adolescents in the United States aged 17 to 21 years [5] suggested that weight and waist circumference were associated with an increased risk of periodontitis. It was found that for each 1 kg increase of weight there was an increase in 6% risk of periodontal disease and for every 1 cm increase in waist circumference, there was a 5% increase in the risk of periodontitis. Further studies suggested that the young adults aged 18 to 34 years with a BMI greater than 30 had a 76% increased risk for periodontal disease, and those with a large waist circumference had a 127% increased risk of disease. It has also been found that periodontitis can be aggravated with obesity especially during adolescence [6,7]. A study done in a Japanese population [6] found that individuals with high BMI and waist-to-hip ratio had a significantly greater risk of developing periodontitis. Another study done in a Jordanian population [8] in 2008, observed that among 340 persons examined, 14% of normal weight subjects had periodontitis, while 29.6% of overweight and 51.9% of obese participants had periodontal disease. Evidence from previous studies suggested that there exists a one-third increase in the prevalence odds of obesity among subjects with periodontal disease [9]. A study done by Hafajee et al. in 2009 suggested that an overgrowth of T. forsythia, one of the pathogenic bacteria belonging to the red complex, was found to be present in the subgingival biofilms of overweight and obese individuals further placing them at higher risk for initiation and progression of periodontal disease. Periodontal disease, being multifactorial in origin, has also been linked to the increase in the levels of certain inflammatory mediators such as interleukins, prostaglandins, and C-reactive protein [10]. It has also been found that these cause the upregulation of endothelin secretions (secreted by endothelial cells) after exposure to pathogenic bacteria and it represents a potent mediator of vascular inflammation and vasoconstriction [10]. Similarly, vitamin D has been found to possess anti-inflammatory and antimicrobial activity, which may be a link for the known interaction of periodontitis and coronary heart disease [11]. Patients with periodontitis and periodontitis with coronary heart disease have been found to have significantly lower serum levels of vitamin D as compared to the healthy controls. Moreover, serum vitamin D levels have been found to be negatively influenced by the presence of periodontitis [11].

The prevalence of obesity in Saudi Arabia has been increasing in this past decade especially among Saudi children and adolescents [12]. An overall higher prevalence (13.4%) of overweight and obesity was found among adolescent girls than boys and these rates were lower in children [13]. However, the variation in the reported prevalence of overweight and obesity could be influenced by a lot of factors including using the Center for Disease Control standards or WHO standards instead of country-specific BMI percentile standards [14]. Overweight and obesity increased significantly with higher levels of socioeconomic status. Similarly, a high prevalence of obesity was found among the adult population of Saudi Arabia. Increasing BMI showed a linear positive association in terms of older age groups among both males and females [15].

The overall prevalence of obesity is estimated to reach up to 59.5% by 2022 [16]. However, no studies regarding the effect of obesity among younger adult age groups on the severity of periodontal disease in the Saudi Arabian population have been reported so far. Hence, the objective of this current study was to assess the prevalence of periodontal disease among obese young adults who visited the outpatient department of Zulfi College of Dental Science, Saudi Arabia. As a secondary aim, we sought to analyze the association between different body mass indexes and the severity of periodontitis among the examined study group. The null hypothesis of this study was that there is no difference in the prevalence of periodontal disease among obese and nonobese young adults in this population.

## 2. Materials and Methods

This descriptive cross-sectional study was designed to assess periodontal disease among obese young adult subjects who visited the outpatient department of Zulfi College of Dental Science between 1st January 2019 to 30th June 2019. The study was approved by the institutional ethical committee of Majmaah University, Saudi Arabia (Research Number: 140/38; dated 19th October 2018) in compliance with the Helsinki Declaration. Four hundred obese patients (Saudi nationals) aged 18–40 years, with BMI 30 and above, recruited by convenience sampling, were screened for eligibility. A total of 329 patients were enrolled for the study, out of which 307 patients gave informed consent to participate. Those excluded were subjects who were had a systemic disease or were on medication for any systemic disease/conditions, patients who received periodontal treatment or antibiotics for at least 3 months before the study, physically and mentally challenged patients, completely edentulous or having any sort of prosthetic appliance, pregnant women or lactating mothers and patients who did not give written consent to participate in the study.

Patients who consented to participate were further stratified into three different age groups (according to the WHO criteria), i.e., 18–20, 21–30, and 31–40 years. Demographic variables that can act as covariants for periodontal disease like age, gender, place of origin (whether from a rural or urban background), level of education (basic education, college or postgraduate), previous dental visits (once a year, 6 months or occasionally), familial history of periodontal disease (present or absent) and dietary patterns (balanced, frequent or irregular) were recorded.

### 2.1. Anthropometric Parameters

BMI was used to quantify obesity, as the WHO recommends that BMI provides the most useful population-level measure of overweight and obesity for both genders. The person’s weight in kilograms was recorded using a standard physician’s scale. The height of the subjects in centimeters was taken using a stadiometer. Body mass index was electronically calculated by dividing the weight by the square of his/her height in meters. Waist circumference (WC) was measured using measuring tape at the midpoint between the lower border of ribs and upper border of pelvis. As per the WHO classification system, BMI patients were classified as obese (BMI ≥30) or nonobese (BMI <30).

### 2.2. Periodontal Parameters

The periodontal status of the participant was assessed according to community periodontal index using a sterilized community periodontal index (CPI) probe and mouth mirror. The Fédération Dentaire Internationale (FDI) system was used for tooth numbering. The dentition was divided into sextants which were defined by tooth numbers −18–14, 13–23, 24–28, 38–34, 33–43 and 44–48. The scores were obtained by examining all teeth and the highest score in each sextant was recorded. The CPI scoring criteria followed were—0 (healthy gingiva), 1 (bleeding on probing), 2 (presence of calculus), 3 (periodontal pockets of 4–5 mm) and 4 (periodontal pockets of ≥6 mm). Those with a CPI score of 3 or more were classified to have ‘presence of periodontal disease’ and those less than 3 were assigned as having stable periodontium. Criteria for scoring loss of attachment (LOA) in the CPI index were: Score 0: healthy periodontal conditions, loss of attachment 0–3, cementoenamel junction (CEJ) not visible, Score 1: loss of attachment 4–5 mm, Score 2: loss of attachment 6–8 mm, Score 3: loss of attachment 9–11 mm, Score 4: loss of attachment 12 mm or more.

Oral Hygiene Index-Simplified (OHI-S) (Greene and Vermillion, 1964) was recorded to assess the oral hygiene status of the individual. Surfaces of six index teeth inspected for OHI-S included labial and buccal surfaces of tooth number 16, 26 11, 31 and lingual surface for 36 and 46. In case 16 was missing or had crowns, then 17 or third molar (18) were examined. In the anterior portion of the mouth, if tooth number 11 and 31were missing then, 21 or 41, respectively, on the opposite side of the midline, were substituted. Scores for both debris and calculus were recorded separately, after which the debris index and calculus index were calculated by adding the individual scores and dividing it by the total number of teeth examined. The OHI-S score for each participant was calculated by adding the debris index and calculus index score. The results were interpreted as good (0–1.2), fair (1.3–3) and poor (3.1–6).

All examiners were trained by the principal investigator before the study and calibrated with volunteers who had features similar to the study population. Examiner calibration was done before the study using a re-examination of 10 volunteers after 2 weeks by the same examiner. The intra-examiner correlation coefficient for repeated measurements was 0.85 (*p* < 0.05), indicating high reliability. Furthermore, clinical measurements were recorded using a double-pass method to minimize measurement errors.

## 3. Data Analysis

The normality tests (Kolmogorov–Smirnov and Shapiro–Wilks tests) results revealed that all variables followed a normal distribution. Therefore, the parametric test was applied to analyze the data. The Chi-Square test was used to assess the association between the sociodemographic variables (all are categorical variables except age) and the CPI index scoring and LOA scoring (both of which are categorical variables).To compare the mean values between the groups, an independent samples t-test was applied. Multivariate binary logistic regression analysis was used to identify the predictors for periodontal disease in the obese young adult population. Analysis of the data was done using SPSS (IBM SPSS Statistics for Windows, Version 20.0. A P-value of less than 0.05 was considered statistically significant.

## 4. Results

Three hundred and twenty-nine adult obese participants who attended the Outpatient Department of Zulfi College of Dental Science from January 2019 to June 2019 were enrolled in the study. Three hundred and seven participants with mean age 28.4 (±7.1) years participated in the study (164 males and 143 females). A total of 22 participants were excluded from the study as they disagreed to sign the consent form. Table 1 summarizes the sociodemographic characteristics, habits, and BMI of the participants. Based on the characteristics of the participants, around 47.9% of the participants were in an age range of 21–30 years. Males were a predominant part of the sample (53.4%). Around half of the participants (54.7%) originated from a rural place and a similar proportion (58.3%) of the participants had graduated from college. Occasional smoking was reported by 31.7% of the participants. About 58% of the participants said they only visited the dentist occasionally and the majority of the study sample (79.5%) reported having no family history of periodontitis. A total of 53.1% of the participants had fair oral hygiene and most of them (71.3%) had CPI score ≥3, indicating the presence of periodontal disease. This table also shows the total distribution of the study subjects along with the distribution of the demographic variables associated with CPI scores <3 and ≥3 in which those with CPI score ≥3 included were 51.6% females, 44.3% between the age of 21–30 and 48.4% aged between 31–40 years, 69.3% who visited the dentist only occasionally and 77.2% had no family history of periodontal disease.

Table 2 shows a comparison between stratified younger adult age and CPI scores for inflammation among obese and extremely obese subjects. It showed an overall significant difference in CPI score for inflammation in terms of BMI (*p* = 0.03). Patients in the 21–30 years of age group had a statistically significant difference in terms of CPI score for inflammation between obese and extremely obese subjects (*p* = 0.05). In Table 3, overall results were found to be similar as the CPI scores on the loss of attachment among obese and extremely obese subjects showed statistically significant results (*p* = 0.01), especially in the 21–30 years of age group (*p* < 0.001). Logistic regression analysis done in Table 4 shows the sociodemographic characters, habits and periodontal parameters concerning periodontal disease. Significant relationships were observed for characteristics such as gender, particular age group, place of origin, visiting the dentist, oral hygiene status, smoking habits, LOA and BMI. No significant differences were noted for the level of education, family history, and dietary habits. Female gender (OR: 0.469; 95%CL: 0.279–0.788; *p* < 0.001), participants between 31–40 years of age (OR: 3.180; 95%CL: 1.337–7.561; *p* < 0.001), urban place of origin (OR: 0.531; 95%CL: 0.321–0.877; *p* < 0.05), participants who last visited a dentist occasionally (OR: 5.965; 95% CL: 3.130–11.368; *p* < 0.001), subjects who smoked >10 cigarettes (OR: 11.868; 95%CL: 3.588–39.254; *p* < 0.001) were found to have significantly higher risk for having chronic periodontitis. Similarly, subjects with fair (OR: 5.391; 95%CL: 2.524–11.513; *p* < 0.001) and poor oral hygiene status (OR: 17.250; 95%CL: 6.958–42.764; *p* < 0.001), LOA of score 2 (OR: 6.172; 95%CL: 0.000–13.123; *p* < 0.001) and BMI ≥40 (OR: 4.458; 95%CL: 0.000–0.9.292; *p* < 0.001) were also found to have a significantly strong risk of having periodontal disease.

## 5. Discussion

In this study, a high prevalence of periodontal disease was observed among obese young adults (71.3%). As we wanted to measure the outcome among the exposed (obesity) at the same point in time, the nonobese population was not included in the study. Recent systematic reviews showed an association between various obesity indicators and periodontitis [17,18,19]. Obesity has been suggested as a strong risk factor for inflammatory periodontal tissue destruction, second only to smoking [20]. Obesity can trigger an immune response by producing cytokines such as tumor necrosis factor α (TNF-α), IL-6, and IL-1, initiating an acute immune response, further suggesting the possibility to induce inflammatory responses inducing a progression of periodontal disease [21]. This, in turn, may induce the infiltration and activation of immune cells, thus creating a pro-inflammatory environment in the body. Available data suggest that increased counts of oral periodontal pathogens, like Tannerella forsythia and Selenomonas noxia, have been associated with obesity [22].

A comparison between various bacterial species in salivary samples of obese and nonobese women revealed that there was a higher percentage of the bacterium Selenomonads noxia (98.4%) in the samples of obese women. This Gram-negative, obligate anaerobic bacterium is likely to produce propionic acid after active fermentation of glucose, further suggesting its biological plausibility in causing periodontal disease. The role of oral bacteria in the development of obesity can be explained by various mechanisms. Oral bacteria may contribute to increased metabolic efficiency, resulting in unwanted weight gain even if there is a small excess in calorie consumption, with no change in diet or exercise. In other instances, the oral bacteria itself could increase appetite following weight gain. Moreover, oral bacteria could alter tissue resistance to insulin by increasing the levels of TNFα or reducing levels of adiponectin [22]. The role of C-reactive protein and fibrinogen produced by the liver is also responsible for the chronic subclinical systemic inflammatory response in obese patients. Furthermore, the lowering of periodontal disease prevalence was found to be associated with increased physical activity and was directly proportional to the frequency of activity [23]. This may be due to the modulatory effects on cytokines production following physical activity [24].

Obese participants who visited the dentist occasionally showed a high prevalence of periodontal disease observed, which is similar to previous results [25] on a Malaysian population. Most of the study sample (79.5%) reported had no family history of periodontitis. This finding was in accordance with the study done by Nabeeh et al. [26] where family history of periodontitis had no association with the occurrence of periodontal disease in a Saudi Arabian population. However, the risk for periodontitis was greater among subjects who reported positive family history in an earlier study by Ababneh et al. [27] in a Jordanian population. Moreover, in our study, subjects with fair and poor oral hygiene status were found to have a significantly stronger risk of having periodontal disease. This finding has been supported by various studies in different populations [28,29,30].

Similar to the earlier reports [31,32,33,34], in this present study, too, obese patients who were smoking >10 cigarettes were at a higher risk of developing periodontal disease. Infrequent visits to the dentist are a risk determinant for periodontal disease in obese patients. Often, young overweight persons visited dentists only for emergency treatment [34,35] which showed that problem-oriented users have poorer oral health. Henceforth, motivating them for regular dental visits could bring about a positive influence in the maintenance of periodontal health. Similarly, in our study, obese patients who visited the dentist infrequently were at high risk for developing periodontitis compared to obese patients who visit the dentist every 6 months.

In this study, the mean weight and BMI of the obese subjects having periodontitis were higher when compared to obese subjects who do not have periodontitis. Similarly, obesity has been associated with an increased occurrence of periodontal disease in other studies [36,37,38]. A recent study reported that a reduction in body weight and BMI has been found to cause a reduction in serum cytokines, such as TNF-alpha, and elevated levels of adiponectin in the serum [39]. Adipose tissue is known to synthesize cytokines and hormones, called adipokines, which play an important role in modulating periodontitis. Iwayama et al. [40] investigated the effect of adiponectin, produced by adipocytes, and found they had a beneficial function in maintaining homeostasis of periodontal health and found that its levels decrease in obese subjects. However, Kim et al. [41] found that no association exists between BMI and periodontal disease in a Korean population, whereas yet another report by Suvan et al. [42] gives inconclusive evidence of the association. The difference in the mean height and waist values was not found to be statistically significant in the present study. Moreover, a significant association was also found between BMI and loss of attachment among the 21–30-year-old study subjects, which is in accordance with the study conducted by Francis et al. [43] AlQahtani et al. reported a direct relationship between obesity and periodontal diseases in young females from Abha, Saudi Arabia as determined by measuring BMI and neck circumferences [44].

Age (OR: 3.180), occasional dental visits (OR: 5.965), smoking >10 cigarettes (OR: 11.868), fair (OR: 5.391), poor oral hygiene status (OR: 17.250), loss of attachment (OR: 6.172) and BMI =/>40 (OR: 4.458) were found to have the most significantly higher risk for having periodontal disease. The findings are similar to previous results [15,30] where these factors have been shown to increase the risk for periodontal disease. Since all the risk factors other than age are modifiable, awareness regarding a healthy lifestyle, introduction of smoking cessation programs, improved oral hygiene methods along with bodyweight management are needed to reduce periodontal disease burdens.

### Strengths and Limitations of the Study

To the best of our knowledge, this is the only study that assessed the incidence and severity of periodontal disease among the obese younger adult age group in the Saudi Arabian population. The differences in these results could be due to heterogeneity in the properties of the study population and parameters chosen to measure obesity (cytokine levels, systemic health of patients, dietary habits, genetic predisposition). Evaluation of subgroups would be possible with a larger sample size. The absence of a nonobese group in the present study is another limitation. A multicenter approach would have been better to increase the generalizability of the results. Being a cross-sectional study, it cannot identify causality between obesity and periodontal disease. Hence, longitudinal studies with more precise measures of adiposity are needed for a better understanding of the relationship between periodontal disease and obesity.

## 6. Future Perspective

This study demonstrates that most obese young adults had periodontal disease, while age, occasional dental visit, smoking >10 cigarettes, and poor oral hygiene were also associated with a significantly higher risk of having periodontal disease. Dental practitioners should educate their younger adult obese patients about the risk of periodontal disease and encourage them to practice and maintain proper oral hygiene. This attempt from the dentist can halt further progression of periodontal disease. The underlying biological mechanisms that link periodontal disease to obesity such as adipose tissue-derived cytokines, tumor necrosis factor-α, interleukin-6 need further evidence-based studies. Hence this research can further help dental professionals to motivate the young adult citizens of Saudi Arabia to maintain normal body weight and adopt a balanced diet and indulge in physical activities for overall wellbeing.

## 7. Conclusions

Within the limitations of this study, it has been found that a high prevalence of periodontal disease is present in obese patients among the Saudi Arabian population. Poor oral hygiene, smoking, loss of attachment, and infrequent dental visits were found as strong predictors of periodontal disease. It is recommended that dental practitioners should educate their obese patients about the risk of periodontal disease and reinforce the importance of proper oral hygiene. Hence, this research can further help dental professionals to motivate the citizens of this country to maintain normal body weight and a healthy lifestyle.

## Figures and Tables

**Table 1 medicina-56-00197-t001:** Distribution of sociodemographic characteristics, habits, and body mass index (BMI) associated with community periodontal index (CPI) scores <3 and ≥3.

Demographic Variable			CPI < 3	CPI ≥ 3	Total
Gender	Male	Count	58	106	164
		%	66.7%	48.2%	53.4%
	Female	Count	29	114	143
		%	33.3%	51.8%	46.6%
Age (years)	18–20	Count	12	16	28
		%	13.8%	7.3%	9.1%
	21–30	Count	50	97	147
		%	57.5%	44.1%	47.9%
	31–40	Count	25	107	132
		%	28.7%	48.6%	43%
Place of residence	Urban	Count	38	130	168
		%	43.7%	59.4%	54.8%
	Rural	Count	49	90	139
		%	56.3%	40.6%	45.2%
Level of Education	Basic	Count	31	83	114
		%	35.6%	37.8%	37.1%
	College	Count	56	123	179
		%	64.4%	55.9%	58.3%
	Postgraduate	Count	0	14	14
		%	0.0%	6.3%	4.6%
Visit to Dentist	Once a year	Count	32	30	62
		%	36.8%	13.6%	20.2%
	6 months	Count	28	37	65
		%	32.2%	16.8%	21.2%
	Occasionally	Count	27	153	180
		%	31.0%	69.6%	58.6%
Family History	Present	Count	12	51	63
		%	13.8%	23.2%	20.5%
	Not present	Count	75	169	244
		%	86.2%	76.8%	79.5%
Diet	Balanced	Count	30	73	103
		%	34.5%	33.2%	33.5%
	Frequent	Count	17	58	75
		%	19.5%	26.4%	24.5%
	Irregular	Count	40	89	129
		%	46.0%	40.4%	42.0%
Oral Hygiene Index-Simplified	Good	Count	27	12	39
		%	31.0%	5.5%	12.7%
	Fair	Count	48	115	163
		%	55.2%	52.2%	53.1%
	Poor	Count	12	93	105
		%	13.8%	42.3%	34.2%
CPI Scores (Loss of attachment)	0	Count	87	15	102
		%	100.0%	6.8%	33.2%
	1	Count	0	116	116
		%	0.0%	52.7%	37.8%
	2	Count	0	89	89
		%	0.0%	40.5%	29%
Smoking	Never	Count	50	134	184
		%	57.50%	60.90%	60.00%
	Occasionally	Count	17	34	51
		%	19.50%	15.40%	16.60%
	<10	Count	13	11	24
		%	14.90%	5.00%	7.80%
	≥10	Count	7	41	48
		%	8.10%	18.70%	15.60%
Body Mass Index (BMI)	30–39	Count	87	209	296
		%	100.0%	95.0%	96.4%
	≥40	Count	0	11	11
		%	0.0%	5.0%	3.6%

**Table 2 medicina-56-00197-t002:** Comparison between stratified younger adult age and CPI score for inflammation among obese and extremely obese subjects.

Age (Years)				CPI-Inflammation		Total	Chi Square Value	*p* Value
				CPI < 3	CPI ≥ 3			
18–20	BMI	30–39	Count	12	14	26	1.615	0.20 ^ns^
			%	46.2%	53.8%	100.0%		
		≥40	Count	0	2	2		
			%	0.0%	100.0%	100.0%		
	Total		Count	12	16	28		
			%	42.9%	57.1%	100.0%		
21–30	BMI	30–39	Count	50	90	140	3.789	0.05 *
			%	35.7%	64.3%	100.0%		
		≥40	Count	0	7	7		
			%	0.0%	100.0%	100.0%		
	Total		Count	50	97	147		
			%	34.0%	66.0%	100.0%		
31–40	BMI	30–39	Count	25	105	130	0.479	0.48 ^ns^
			%	19.2%	80.8%	100.0%		
		≥40	Count	0	2	2		
			%	0.0%	100.0%	100.0%		
	Total		Count	25	107	132		
			%	18.9%	81.1%	100.0%		
Total	BMI	30–39	Count	87	209	296	4.533	0.03 *
			%	29.4%	70.6%	100.0%		
		≥40	Count	0	11	11		
			%	0.0%	100.0%	100.0%		
	Total		Count	87	220	307		
			%	28.4%	71.6%	100.0%		

ns: not significant; * statistically significant at *p* value < 0.05

**Table 3 medicina-56-00197-t003:** Comparison between stratified younger adult age and CPI scores on loss of attachment among obese and extremely obese subjects.

Age (Years)				CPI-Loss of Attachment			Total	Chi Square Value	*p* Value
				0	1	2			
18–20	BMI	30–39	Count	12	14	0	26	1.615	0.20 ^ns^
			%	46.2%	53.8%	0	100.0%		
		≥40	Count	0	2	0	2		
			%	0.0%	100.0%	0	100.0%		
	Total		Count	12	16	0	28		
			%	42.9%	57.1%	0	100.0%		
21–30	BMI	30–39	Count	62	44	34	140	19.002	0.001 **
			%	44.3%	31.4%	24.3%	100.0%		
		≥40	Count	0	0	7	7		
			%	0.0%	0.0%	100.0%	100.0%		
	Total		Count	62	44	41	147		
			%	42.2%	29.9%	27.9%	100.0%		
31–40	BMI	30–39	Count	28	54	48	130	2.720	0.25 ^ns^
			%	21.5%	41.6%	36.9%	100.0%		
		≥40	Count	0	2	0	2		
			%	0.0%	100.0%	0.0%	100.0%		
	Total		Count	28	56	48	132		
			%	21.2%	42.8%	36.4%	100.0%		
Total	BMI	30–39	Count	102	112	82	296	8.637	0.01 *
			%	34.5%	37.8%	27.7%	100.0%		
		≥40	Count	0	4	7	11		
			%	0.0%	36.4%	63.6%	100.0%		
	Total		Count	102	116	89	307		
			%	33.3%	37.8%	28.9%	100.0%		

ns: not significant; * statistically significant at *p* value < 0.05; ** statistically significant at *p* value < 0.001.

**Table 4 medicina-56-00197-t004:** Logistic regression showing sociodemographic characters, habits and periodontal parameters in relation to periodontal disease.

Predictors		CP OR (Odds Ratio)	95% CI		*p* Value
			Lower	Upper	
Gender	Male	1 (Reference)			
	Female	0.469	0.279	0.788	0.004 **
Age (years)	18–20	1 (Reference)			
	21–30	1.455	0.639	3.312	0.372 ^ns^
	31–40	3.180	1.337	7.561	0.009 **
Place of residence	Rural	1 (Reference)			
	Urban	0.531	0.321	0.877	0.013 *
Education	Basic	1 (Reference)			
	College	0.851	0.505	1.433	0.544 ^ns^
	Post grad	0.10	0.000	1.111	0.998 ^ns^
Visit to the dentist	Once a year	1 (Reference)			
	6 months	1.410	0.700	2.837	0.336 ^ns^
	Occasionally	5.965	3.130	11.368	0.001 **
Family History	Yes	1 (Reference)			
	No	0.541	0.272	1.074	0.079 ^ns^
Diet	Balanced	1 (Reference)			
	Frequent	1.422	0.714	2.829	0.316 ^ns^
	Irregular	0.927	0.526	1.633	0.793 ^ns^
OHI-S	Good	1 (Reference)			
	Fair	5.391	2.524	11.513	0.000 **
	Poor	17.250	6.958	42.764	0.000 **
CPI Scores (Loss of attachment)	0	1 (Reference)			
	1	2.197	0.000	3.835	0.495 ^ns^
	2	6.172	0.000	13.123	0.000 **
Smoking	Never	1 (Reference)			
	Occasionally	2.533	1.080	5.941	0.033 *
	<10	1.368	0.498	3.760	0.543 ^ns^
	>10	11.868	3.588	39.254	0.001 **
BMI	30–39	1 (Reference)			
	=/>40	4.458	0.000	9.292	0.000 **

Multivariate binary logistic regression; ns: not significant; * statistically significant at *p* value < 0.05; ** statistically significant at *p* value < 0.001; CL: Confidence Interval.

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
