# Peer review of "Prevalence of Periodontal Disease among Obese Young Adult Population in Saudi Arabia—A Cross-Sectional Study"

_medicina, 2020, doi:10.3390/medicina56040197_

Round 1
Reviewer 1 Report
The reviewer assumes that Chi-Square-tests were used instead of t-tests as the authors categorized the metric variables / values in groups.
Thus, the "Analysis"-paragraph should be corrected.
Author Response
Reviewer-1
- The reviewer assumes that Chi-Square-tests were used instead of t-tests as the authors categorized the metric variables/values in groups. Thus, the "Analysis"-paragraph should be corrected.
Authors’ response: The authors would like to thank you for taking the time to review our manuscript. We appreciate the constructive comments.
Chi-Square test was used to assess the association between the sociodemographic variables (all are categorical variables except Age) and the CPI index scoring and LOA scoring (both of which are categorical variables).
Please note that the analysis paragraph has been corrected as follows-
The Normality tests Kolmogorov-Smirnov and Shapiro-Wilks test results reveal that all variables follow Normal distribution. Therefore, the parametric test was applied to analyze the data. Chi-Square test was used to assess the association between the sociodemographic variables (all are categorical variables except age) and the CPI index scoring and LOA scoring (both of which are categorical variables).To compare the mean values between the groups, an independent samples t-test was applied. Multivariate binary logistic regression analysis was used to identify the predictors for periodontal disease in the obese young adult population. Analysis of the data was done using SPSS (IBM SPSS Statistics for Windows, Version 20.0. A P-value of less than 0.05 was considered statistically significant.
Reviewer 2 Report
Thank you for your paper. The manuscript focuses on evaluating the incidence of periodontal disease in Saudi obese young adults.
The major limitation of the study is that it lacks the control group (non-obese participants). The authors should provide more epidemiological data on prevalence of obesity in Saudi Arabia in the specific age group.
The study does not explain the coincidence of obesity and periodontal disease. Moreover, it proved that there were no significant differences noted for the level of education, family history, dietary habits, and BMI. Given this result, the lack of study group seems to be a compelling issue. The results of the study were not discussed enough. The study proved the obvious, that poor oral hygiene status links to a significantly strong risk of having periodontal disease.
The manuscript needs some adjustments to the editorials requirements.
Some statements need clarifications i.e.:
line 24: “younger adults”
lines 25-27: “Patients between 21-30 years of age had a statistically significant difference in terms of CPI score for inflammation and loss of attachment among obese and extreme obese
subjects showed statistically significant results (p<0.05 and p<0.001 respectively)
line 42: “younger middle-aged” so younger or middle aged? please define
Please, define “visitors of Outpatient Department”. Were they patients? It would be preferable to call them “subjects” or “participants”.
The number of participants do not add up. First the authors mention (line 75) that “329 patients were enrolled for the study and gave informed consent to participate.”
Later on, it is said that (line 125) “Three hundred and six consecutive new adult obese patients (164 males and 143 females) who attended the Outpatient Department of Zulfi College of Dental Science from January 2019 to June 126 2019 were enrolled in the study.”
line 127: “Three hundred and seven participants..”
The statement in line 128: “22 patients did not complete the study” is confusing. Were these 22 within 307 subjects?
Which is it? Please explain.
The results should be presented more clearly. Table 1 misses BMI distribution among study subjects.
The statements in paragraph 146-161 should be clarified:
“patients… had a statistically significant difference..” how were they different?, “overall results were found to be similar…” to what?
lines 153-154: “No significant differences were noted for the level of education, family history, dietary habits, and BMI.”
line 159: what does “CP” stand for?
lines 159-161: “Similarly, subjects with fair (OR:5.391;95%CL:2.524-11.513; p<0.001) and poor oral hygiene status (OR:17.250; 95%CL:6.958-42.764; p<0.001) were also found to have a significantly strong risk of having periodontal disease.” Well, that seemed to be obvious even before the study.
Line 206 (“In this study, the mean weight and BMI of the study subjects having periodontitis were higher when compared to the control group.”) mentions the presence of the control group in this study. What was the control group? No information is included in the Materials and Method section. Moreover, the authors stated in line 232 that the absence of a non-obese group (as control group??) in the present study is another limitation of the study.
Please, correct the statement in line 227 (”the effect of obesity among younger adult age groups on the severity of periodontal disease..”) as the study did not investigated the effect, only the incidence of periodontal disease in obese patients.
line 241: “younger obese patients” meaning?
line 245: “this country” which one?
some grammar mistakes:
line 67: differing or different
line 73: starting the sentence with numeral
line 76: “who had any”
lines 88-89: missing a verb
Author Response
Reviewer-2
Thank you for your paper. The manuscript focuses on evaluating the incidence of periodontal disease in Saudi obese young adults.
- The major limitation of the study is that it lacks the control group (non-obese participants). The authors should provide more epidemiological data on the prevalence of obesity in Saudi Arabia in the specific age group.
Authors’ response:
Thank you for your valuable suggestion. The authors would like to express our gratitude for taking the time to review our manuscript. We appreciate the constructive comments.
The aim of our study was to assess the prevalence of periodontal disease among obese young adults and to see whether the severity of the periodontal disease varies with the BMI scores. In this study, a high prevalence of periodontal disease was observed among obese young adults (71.3%). As we wanted to measure the outcome among the exposed (obesity) at the same point in time, non-obese population was not included in the study.
Please note that we have added more epidemiological data on the prevalence of obesity in Saudi Arabia.
- The study does not explain the coincidence of obesity and periodontal disease. Moreover, it proved that there were no significant differences noted for the level of education, family history, dietary habits, and BMI. Given this result, the lack of study group seems to be a compelling issue. The results of the study were not discussed enough. The study proved the obvious, that poor oral hygiene status links to a significantly strong risk of having periodontal disease.
Authors’ response:
Thank you for this comment. Yes, we agree that poor oral hygiene status is already regarded to be a significant risk factor in the etiology of periodontal disease. But there are many modifying factors that can aggravate the severity of the periodontal disease. The biological plausibility of obesity as the causative factor of periodontal disease is added to the discussion in the manuscript.
A comparison between the various bacterial species in salivary samples of obese and non-obese women revealed that there was a higher percentage of bacterium Selenomonads noxia (98.4%) in the samples of obese women. This gram-negative, obligate anaerobic bacteria is likely to produce propionic acid after active fermentation of glucose, further suggesting its biological plausibility in causing periodontal disease. The role of oral bacteria in the development of obesity can be explained by various mechanisms. Oral bacteria may contribute to increased metabolic efficiency, resulting in unwanted weight gain even if there is small excess in calorie consumption, with no change in diet or exercise. In other instances, the oral bacteria itself could increase appetite following weight gain. Moreover, oral bacteria could alter the tissue resistance to insulin by increasing the levels of TNFα or reducing levels of adiponectin.
- The manuscript needs some adjustments to the editorials requirements.
Authors’ response: Thank you for your valuable suggestion.
Please note that we have made all the possible corrections and modifications.
- Some statements need clarifications i.e.:
line 24: “younger adults”
Authors’ response:
We have changed it to “Young obese adult groups of age between 18 – 40 years”
- lines 25-27: “Patients between 21-30 years of age had a statistically significant difference in terms of CPI score for inflammation and loss of attachment among obese and extreme obese.
Authors’ response:
Please note that Patients in the 21-30 years of age group had a statistically significant difference in terms of CPI score for inflammation between obese and extreme obese subjects (p=0.05). Overall results were found to be similar as the CPI scores on the loss of attachment among obese and extreme obese subjects showed statistically significant results (p=0.01), especially in the 21-30 years of age group (p<0.001).
- subjects showed statistically significant results (p<0.05 and p<0.001 respectively)
Authors’ response:
Please note that Patients in the 21-30 years of age group had a statistically significant difference in terms of CPI score for inflammation between obese and extreme obese subjects (p=0.05). Overall results were found to be similar as the CPI scores on the loss of attachment among obese and extreme obese subjects showed statistically significant results (p=0.01), especially in the 21-30 years of age group (p<0.001).
- line 42: “younger middle-aged” so younger or middle aged? please define
Authors’ response:
Kindly note that we have corrected it to young obese adults where the mean age of the population was 22.40 years.
- Please, define “visitors of Outpatient Department”. Were they patients? It would be preferable to call them “subjects” or “participants”.
Authors’ response:
Kindly note that this has been changed. The word visitors has been changed to subjects; subjects who visited the outpatient department of Zulfi College of Dental Science
- The number of participants do not add up. First the authors mention (line 75) that “329 patients were enrolled for the study and gave informed consent to participate.”
Later on, it is said that (line 125) “Three hundred and six consecutive new adult obese patients (164 males and 143 females) who attended the Outpatient Department of Zulfi College of Dental Science from January 2019 to June 126 2019 were enrolled in the study.”
Authors’ response:
We would like to state that 329 patients were enrolled for the study out of which only 307 patients gave informed consent to participate in the study. This has been corrected in the manuscript.
- line 127: “Three hundred and seven participants.”
Authors’ response:
This error has been corrected. Three hundred and twenty-nine adult obese participants who attended the Outpatient Department of Zulfi College of Dental Science from January 2019 to June 2019 were enrolled in the study. Three hundred and seven participants with mean age 28.4 (±7.1) participated in the study (164 males and 143 females). 22 participants were excluded from the study as they disagreed to sign the consent form.
- The statement in line 128: “22 patients did not complete the study” is confusing. Were these 22 within 307 subjects? Which is it? Please explain.
Authors’ response:
This error has been corrected. Three hundred and twenty-nine adult obese participants who attended the Outpatient Department of Zulfi College of Dental Science from January 2019 to June 2019 were enrolled in the study. Three hundred and seven participants with mean age 28.4 (±7.1) participated in the study (164 males and 143 females). 22 participants were excluded from the study as they disagreed to sign the consent form.
- The results should be presented more clearly. Table 1 misses BMI distribution among study subjects.
Authors’ response:
Please note that Table 1 and 2 has been combined.
- The statements in paragraph 146-161 should be clarified:
Authors’ response: This has been corrected in the manuscript.
- “patients… had a statistically significant difference.” how were they different? “overall results were found to be similar…” to what?
Authors’ response: This has been corrected in the manuscript.
- lines 153-154: “No significant differences were noted for the level of education, family history, dietary habits, and BMI.”
Authors’ response: We would like to inform that we ran the regression analysis once again and found that the values for BMI are now significant. This change has been added in the result section.
- line 159: what does “CP” stand for?
Authors’ response:
CP stands for Chronic periodontitis. The change has been made in the manuscript.
- lines 159-161: “Similarly, subjects with fair (OR:5.391;95%CL:2.524-11.513; p<0.001) and poor oral hygiene status (OR:17.250; 95%CL:6.958-42.764; p<0.001) were also found to have a significantly strong risk of having periodontal disease.” Well, that seemed to be obvious even before the study.
Authors’ response:
Yes, there has been already established relationship between oral hygiene and periodontal disease, the results of the study is also consistently stating the same, which is in accordance with the previous studies.
- Line 206 (“In this study, the mean weight and BMI of the study subjects having periodontitis were higher when compared to the control group.”) mentions the presence of the control group in this study. What was the control group? No information is included in the Materials and Method section. Moreover, the authors stated in line 232 that the absence of a non-obese group (as control group??) in the present study is another limitation of the study.
Authors’ response:
Thank you for your valuable suggestion. Yes, we agree, have corrected the error in the manuscript. In this study, the mean weight and BMI of the study obese subjects having periodontitis were higher when compared to the obese subjects who does not have periodontitis
- Please, correct the statement in line 227 (”the effect of obesity among younger adult age groups on the severity of periodontal disease..”) as the study did not investigated the effect, only the incidence of periodontal disease in obese patients.
Authors’ response:
Thank you for the suggestion. We have corrected this in the manuscript. It now reads - which assessed the incidence and severity of periodontal disease among obese younger adult age group in Saudi Arabian population.
- line 241: “younger obese patients” meaning?
Authors’ response:
This has been changed to -Young obese adult age groups between 18 -40 years
- line 245: “this country” which one?
Authors’ response:
This change has been made in the manuscript.
Professionals to motivate the young adult citizens of Saudi Arabia to maintain normal body weight and adopt a balanced diet and indulge in physical activities for overall wellbeing.
- some grammar mistakes: line 67: differing or different
Authors’ response:
This change has been made in the manuscript.
Changed from differing to different
- line 73: starting the sentence with numeral
Authors’ response:
This change has been made in the manuscript.
400 has been made into four hundred.
- line 76: “who had any”
Authors’ response:
This change has been made in the manuscript.
Changed to - who were having systemic disease
- lines 88-89: missing a verb
Authors’ response:
Thank you for your valuable suggestion. This change has been made in the manuscript.
BMI was used to quantify obesity as WHO recommends that BMI provides the most useful population-level measure of overweight and obesity for both genders.
Reviewer 3 Report
There are multiple errors of capitalization and punctuation, as well as some awkwardly worded sentences. Please have someone review and correct these.
BMI = weight divided by the height squared. This is not the same as kg/sq.m.
Introduction- should contain a description the biological mechanism that is suspected in the relationship between obesity and periodontal disease. Some of this information is in the discussion-should be moved to the intro.
You mention rate of obesity in page 2 line 61. You mean prevalence.
Materials and Methods- The age range of interested is described using different words in different areas of the text-be consistent.
Each variable reported should be defined. For example, history of periodontal disease, is this self report? Same for dietary and behavior patterns.
What type of dental numbering system is used for the calculation of CPI? You need to specify.
A control group is mentioned. There is not control group in a cross sectional study.
Different systems to note references are used. Please be consistent.
Data Analysis- This section is missing information on logistic regression that will be conducted.
Results- BMI is not included in Table 1 although the text and table heading say that it is.
Tables 1 and 2 can probably be combined.
In methods CPI (inflamation) and CPI (loss of attachment) are not defined clearly, yet these are your outcomes.
Row headings should be more specific.
You should justify whey the BMI greater than 40 group was looked at separately since it is such a small number.
Your logistic regression results are not presented in the manner that is common. Which were the control variables, and when controlled, what is the influence of obesity CPI?
Author Response
Reviewer-3
- There are multiple errors of capitalization and punctuation, as well as some awkwardly worded sentences. Please have someone review and correct these.
Authors’ response: The authors would like to thank you for taking the time to review our manuscript. We appreciate the constructive comments. Kindly note that all these changes have been made in the manuscript.
- BMI = weight divided by the height squared. This is not the same as kg/sq.m.
Authors’ response: Please note that we have used the following method to measure BMI.
BMI was used to quantify obesity as WHO recommends that BMI provides the most useful population-level measure of overweight and obesity for both genders. The person's weight in kilograms using a standard physician's scale. The height of the subject's centimeters was taken using a stadiometer. Body Mass index was electronically calculated by dividing the weight by the square of his/her height in meters.
- Introduction- should contain a description the biological mechanism that is suspected in the relationship between obesity and periodontal disease. Some of this information is in the discussion-should be moved to the intro.
Authors’ response:
We have added the biological mechanism in the introduction section.
- You mention rate of obesity in page 2 line 61. You mean prevalence.
Authors’ response:
Yes, we mean prevalence. This has been corrected.
Overall prevalence of obesity is estimated to reach up to 59.5% by 2022.
- Materials and Methods- The age range of interested is described using different words in different areas of the text-be consistent.
Authors’ response:
We have corrected the error in the manuscript.
- Each variable reported should be defined. For example, history of periodontal disease, is this self report? Same for dietary and behaviour patterns.
Authors’ response: Thank you. We have corrected this in the manuscript.
- What type of dental numbering system is used for the calculation of CPI? You need to specify.
Authors’ response:
This has been now specified in the manuscript. The Fédération Dentaire Internationale (FDI) system was used for tooth numbering. The dentition was divided into sextants which were defined by tooth numbers – 18-14, 13-23,24-28,38-34,33-43 and 44-48.
- A control group is mentioned. There is not control group in a cross-sectional study.
Authors’ response-
This has been rectified in the manuscript.
- Different systems to note references are used. Please be consistent.
Authors’ response:
We have used vancouver style of referencing.
- Data Analysis- This section is missing information on logistic regression that will be conducted.
Authors’ response:
The mistake has been rectified and changes have been made in the analysis section.
- Results- BMI is not included in Table 1 although the text and table heading say that it is.
Authors’ response:
Table 1 and 2 have been combined and a suitable heading has been given.
- Tables 1 and 2 can probably be combined.
Authors’ response:
These tables have been combined
- In methods CPI (inflamation) and CPI (loss of attachment) are not defined clearly, yet these are your outcomes.
Authors’ response:
CPI bleeding on probing is considered as early sign of inflammation which is specified in the scoring criteria
CPI scoring criteria for loss of attachment is added in the manuscript.
- Row headings should be more specific.
Authors’ response:
The changes have been made.
- You should justify whey the BMI greater than 40 group was looked at separately since it is such a small number.
Authors’ response:
The sample size greater than 40 was small is one of the limitation in our study, however it was not related to statistical analysis.
- Your logistic regression results are not presented in the manner that is common. Which were the control variables, and when controlled, what is the influence of obesity CPI?
Authors’ response:
Thank you for your valuable suggestion. Binary logistic regression was used, for example when we conducted the association between BMI and CP all the other independent variables were added as covariates / control variables. (Table prepared keeping the Key article as reference; Khan et al. Prevalence of chronic periodontitis in an obese population: a preliminary study BMC Oral Health (2015) 15:114
Round 2
Reviewer 2 Report
Thank you for revised manuscript and for addressing reviewer's previous comments. The authors significantly improved the manuscript.
Minor issue:
line 30: “On comparing the CPI 30 score for inflammation and LOA, a statistically significant difference was found between the obese and extreme-obese patients between 21-30 years of age (p<0.05 and p<0.001 respectively). ”
I would specify, for which group the indices were statistically significantly higher or lower.
Author Response
Obese and extreme-obese patients together showed a statistically significant difference in the age group of 21-30 years in terms of CPI score for inflammation (p<0.05) and LOA (p<0.001).
Thank you.